# Underwater Topography Inversion in Liaodong Shoal Based on GRU Deep Learning Model

**Zihao Leng [1], Jie Zhang [2,3], Yi Ma [2,\*] and Jingyu Zhang [2]**

[1]  School of Geosciences, China University of Petroleum (East China), Qingdao 266580, China; b19010073@s.upc.edu.cn

[2]  First Institute of Oceanography, Ministry of Natural Resources, Qingdao 266061, China; zhangjie@fio.org.cn (J.Z.); zhangjingyu@fio.org.cn (J.Z.)

[3]  College of Oceanography and Space Informatics, China University of Petroleum (East China), Qingdao 266580, China

\*  Correspondence: mayimail@fio.org.cn; Tel.: +86-532-8896-7094

**Abstract:** The Liaodong Shoal in the east of the Bohai Sea has obvious water depth variation. The clear shallow water area and deep turbid area coexist, which is characterized by complex submarine topography. The traditional semi-theoretical and semi-empirical models are often difficult to provide optimal inversion results. In this paper, based on the traditional principle of water depth inversion in shallow areas, a new framework is proposed in combination with the deep turbid sea area. This new framework extends the application of traditional optical water depth inversion methods, can meet the needs of the depth inversion work in the composite sea environment. Moreover, the gate recurrent unit (GRU) deep-learning model is introduced to approximate the unified inversion model by numerical calculation. In this paper, based on the above-mentioned inversion framework, the water depth inversion work is processed by using the wide range images of GF-1 satellite, then the relevant analysis and accuracy evaluation are carried out. The results show that: (1) for the overall water depth inversion, the determination coefficient $R^2$ is higher than 0.9 and the MRE is lower than 20% are obtained, and the evaluation index shows that the GRU model can better retrieve the underwater topography of this region. (2) Compared with the traditional log-linear model, Stumpf model, and multi-layer feedforward neural network, the GRU model was significantly improved in various evaluation indices. (3) The model has the best inversion performance in the 24–32 m-depth section, with a MRE of about 4% and a MAE of about 1.42 m, which is more suitable for the inversion work in the comparative section area. (4) The inversion diagram indicates that this model can well reflect the regional seabed characteristics of multiple radial sand ridges, and the overall inversion result is excellent and practical.

**Keywords:** deep learning; gate recurrent unit (GRU); water depth inversion

## 1. Introduction

Water depth is an important element for marine scientific research, transportation and shipping, resource development, engineering construction, and environmental protection. Compared with the traditional on-site measurement technology, the use of satellite remote sensing to measure water depth has the advantages of large spatial coverage, low cost, and repeatable observation. It is especially suitable for the inversion of shallow water bathymetry where ships are difficult to enter. It is convenient for making bathymetric maps in large-scale sea areas and makes up for the deficiencies of field bathymetric survey to a certain extent.

Since the 1960s, with the vigorous development of multispectral and hyperspectral satellite remote sensing technology, water depth optical detection technology has attracted wide attention from

relevant scholars [1], and the method of water depth remote sensing inversion model has also been rapidly developing, mainly in three different approaches: theoretical analytical model, semi-theoretical and semi-empirical models and statistical model [2]. Theoretical analytical models are based on the radiative transfer method of the water field, and an expression with the radiance of the remote sensor entrance and bottom material reflection is used to calculate the water depth. Many scholars have made great efforts to establish various theoretical analytical models [3–6], and these models usually have high accuracy and clear physical meaning. However, such models require many water optical parameters, complex to calculate and difficult to obtain, which also limits the application of these water depth inversion methods. Based on the combination of theoretical model and empirical parameters, semi-theoretical and semi-empirical models greatly reduce the computational complexity of inversion with the premise of ensuring a certain universality and inversion accuracy. They are also the most widely used models for water depth optical remote sensing. Among them, the log-linear model [7] is most widely used, Paredes et al. [8] further proposed the dual-band log-linear model, and Stumpf et al. [9] proposed the logarithmic conversion ratio model, which is commonly known as the Stumpf model. Statistical models directly establish the statistical relationship between the radiance value of remote sensing image and the measured water depth; common models include the power function model, logarithmic function model, and linear model [10–13]. They do not consider the physical mechanism of water depth remote sensing, but directly seek the mathematical relationship between water depth and image radiance value, at a specific time, and suitable sea areas also have considerable inversion capabilities.

In recent years, scholars have made much progress in the field of shallow water depth inversion. Kerr et al. [14] develop an approach for predicting water depth in tropical carbonate landscapes from a multispectral satellite image without the need for ground-truth data. Goodman et al. [15] utilized hyperspectral data to evaluate the performance and sensitivity of a representative semi-analytical inversion model for deriving water depth and benthic surface reflectance. With the development of airborne LiDAR, a series of bathymetry research with higher accuracy has been carried out [16–18]. With the development and application of machine learning, especially deep learning models, increasing scholars are beginning to apply machine-learning methods in water depth inversion research. Manessa et al. [19] applied random forest (RF) regression to estimate the water depth of shallow coral reefs, Wang et al. [20] used a spatial distribution support vector machine (SVM) model to perform water depth inversion research and achieved high precision. Multi-layer perceptron (MLP) neural network water depth inversion [21,22] is a special form of a statistical model. On the premise of sufficient training samples, it usually has a better adaptability and higher inversion accuracy than the traditional statistical method. As one of the most classical models in deep learning, convolutional neural network (CNN) models have also been successfully applied to remote sensing image processing [23].

The above-mentioned water depth optical remote sensing inversion models express the relationship between the reflected light information of the seabed and the sea water depth and has been widely used in waterway engineering and reef detection [24–28]. However, they are only applicable to shallow sea areas, and the inversion effect depends on the penetration ability of sunlight into the water body. In the water body with comparatively deep water or high light attenuation coefficient, it is difficult for sunlight to directly penetrate the water body and reflect the bottom reflection information to the remote sensor [29]. Therefore, the model cannot effectively describe the depth information of this kind of sea area, which restricts the development of optical remote sensing depth detection in relatively deep areas. Previous studies have confirmed that the trend of seabed topography will have a regular impact on the water flowing under the sea surface, and the water flow changes further modulate the distribution of micro-scale waves on the sea surface [30], resulting in changes in the distribution of micro-scale waves on the sea surface. After the process of light reflection and scattering, the changes of water topography are shown in the remote sensing images by different brightness degrees [31]. The above-mentioned mechanism explains that it is possible to visually distinguish changes in the water depth of relatively deep areas from remote sensing images, such as the Taiwan Shoal with a water depth of 0–35 m [32],

and the Liaodong Shoal with a water depth of 0–32 m. However, practical applications often require remote sensing models that consider both shallow and deep water. Shallow water and deep water often coexist in a certain sea area. Because light reflected from the seabed is often difficult to capture by remote sensors directly in deep areas, it is not appropriate to directly apply the inversion methods for shallow areas in these areas. This paper attempts to propose a new water depth inversion framework, which can uniformly adapt to the above two kinds of water depth optical imaging mechanisms and can be applied to the sea water depth optical remote sensing inversion in shallow or relatively deep areas at the same time.

The gate recurrent unit network (GRU) [33] is a typical model in the field of deep learning; it is essentially a special artificial neural network with self-connections inside. It is proposed to solve the problems of gradient vanishing, and gradient exploding in the general RNN model, and accurately model the data with short-term or long-term dependence. GRU can also be regarded as a variant of the classic RNN model long short-term memory (LSTM), which can achieve competitive performance as LSTM with less computing resources. At the same time, the GRU model needs less training parameters, so it is more suitable for the water depth inversion problem with usually less training data. Given its excellent learning ability, GRU has been widely used in many fields, such as speech recognition [34], machine translation [35], medical research [36], etc. For water depth inversion in sea areas where shallow water and deep water coexist, it can be viewed from the perspective of piecewise function; that is, the function is composed of shallow and deep water inversion models simultaneously. However, due to the unknown spatial range of shallow and deep areas, it is difficult to accurately define the definition range of the piecewise function. This paper considers the use of the GRU deep learning method to regress this complex piecewise function uniformly, that is, to express the depth of shallow water and deep water at the same time. This model can effectively learn the abundant spectral dimension sequence features of multispectral remote sensing images and establish the complex mapping relationship between the spectral features of remote sensing images and sea depth values.

In this paper, a new unified depth inversion framework is proposed based on the GRU model by using the GF-1 wide-field view (WFV) data covering the Liaodong Shoal and 1:150,000 scale sea chart depth data. Aiming at the complex sea areas where relatively deep areas and shallow areas coexist, a complex mapping relationship between spectral features of remote sensing image and water depth value in different sections is established. For the relatively deep area, by analyzing the relationship among seabed information, sea surface micro-scale waves, and remote sensing image values, the feasibility of passive optical remote sensing water depth inversion in comparatively deep areas is analyzed. Finally, comparative experiments are designed with the log-linear model, Stumpf model and other traditional methods, the experiments are carried out from the overall and segment aspects, and the correlation analysis and accuracy evaluation are followed.

## 2. Data and Research Area

### 2.1. Research Area

The Liaodong Shoal is located in the eastern part of the Bohai Sea, which lies in the south of the Liaodong Bay and north of the Laotieshan Waterway (Figure 1). It is adjacent to the east of the Liaodong Peninsula and belongs to the south extension part of the Liaodong Bay. This shoal is a unique topography of the Liaodong Bay. The main feature is six radial fingerlike sand ridges, and it has the most obvious variation of water depth in Liaodong Bay.

### 2.2. Datasets

The remote sensing image used in this paper comes from the Chinese GF-1 satellite, which is the first satellite of China's high-resolution earth observation system. It has the characteristics of combining high and medium spatial resolution earth observation with a wide range of imaging. The band parameters and some satellite metadata are shown in Table 1.

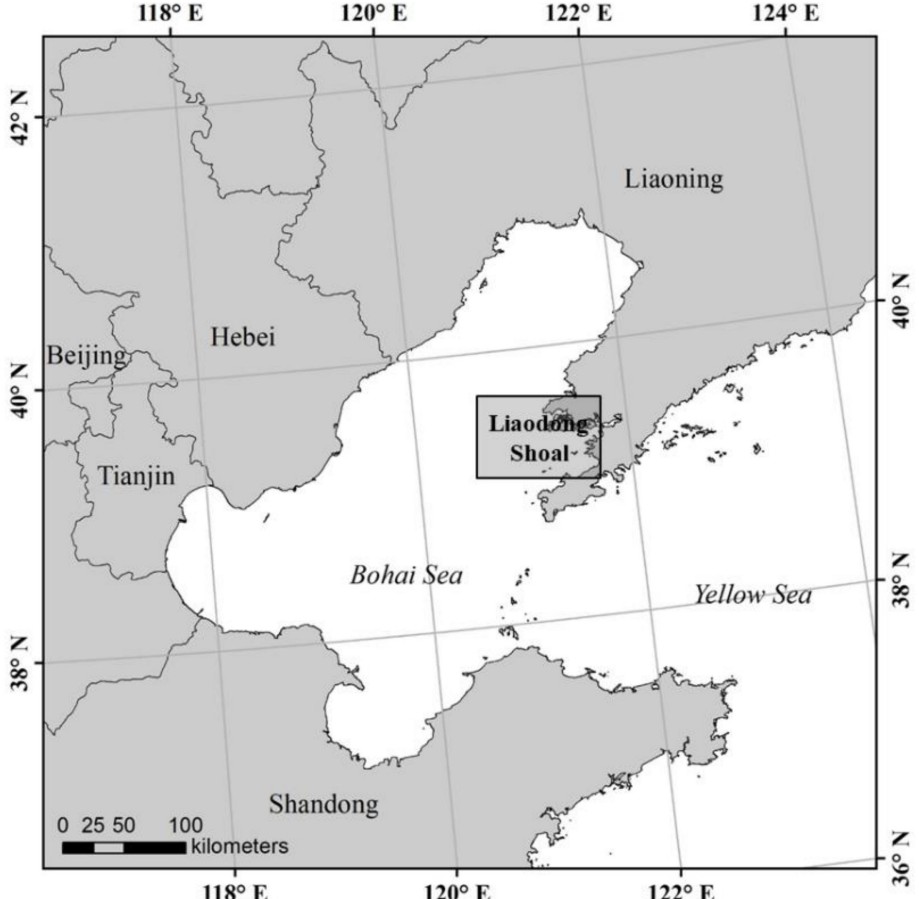

**Figure 1.** Position of Liaodong Shoal.

**Table 1.** GF-1 wide-field view (WFV) image parameters.

| Parameters | Value |
|---|---|
| Spectral region | Band 1:450–520 nm |
|  | Band 2:520–590 nm |
|  | Band 3:630–690 nm |
|  | Band 4:770–890 nm |
| Spatial resolution | 16 m |
| Swath | 800 km |
| Solar azimuth | 159.4° |
| Solar zenith | 56.1° |
| Satellite azimuth | 101.4° |
| Satellite zenith | 63.3° |

This paper uses the four-band data of the GF-1 multispectral image (Figure 2a) referenced to the WGS-84 coordinate system, acquired on 8 April 2016, at 03:11:58 (UTC) with a spatial resolution of 16 m, and a size of 4300 × 5500 pixels. To facilitate the specific grasp of the experimental area, the 1:150,000 scale chart of Dalian Port to Changzuizi in 2005 is introduced here, which is shown in Figure 2b. The points used for training the water depth inversion model were collected from this chart with ArcGIS software.

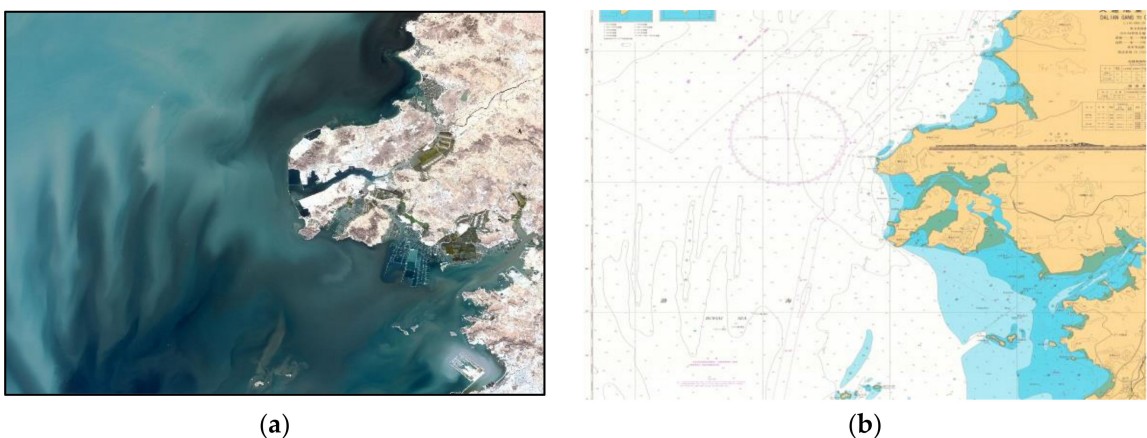

(**a**)                                                                    (**b**)

**Figure 2.** (**a**) GF-1 remote sensing image of Liaodong Shoal; (**b**) sea chart of Liaodong Shoal.

*2.3. Data Preprocessing*

2.3.1. Radiance Conversion

The signal received by the remote sensor is represented as a dimensionless digital, and before quantitative remote sensing research, numerical conversion processing is required. Radiance conversion [37] is to convert the dimensionless *DN* value recorded by the remote sensor into the radiance value or reflectivity of the top atmospheric layer with practical significance. The specific formula is as follows:

$$L = Gain * DN + Bias \tag{1}$$

where *L* is the radiance value of the remote sensor's pupil; *DN* is the sensor observation value, that is, the gray value of the image; *Gain* and *Bias* are the image gain value and bias value respectively, both of which can be obtained from the metadata file of the GF-1 remote sensing image.

2.3.2. Atmospheric Correction

During the transmission process of sunlight in the atmosphere, it will have effects such as reflection and refraction with particles of different sizes in the air, which will interfere with the image imaging, thus resulting in the deviation of the obtained surface reflectance. Therefore, atmospheric correction for the image is needed. In this experiment, the fast line-of-sight atmospheric analysis of spectral hypercubes (FLAASH) method [38] is used to correct the atmosphere of the GF-1 remote sensing image. After atmospheric correction, the influence of Rayleigh scattering in short waves (mainly refers to the blue wave segment) can be reduced

2.3.3. Geometric Correction

Geometric correction [37] includes orthorectification of the image and geometric correction of the sea chart.

(1)   Orthorectification: refers to the aid of the digital elevation model (DEM). Each pixel in the image is corrected to make the image meet the requirements of orthographic projection. The purpose is to eliminate the influence of topography or the deformation caused by the orientation of the camera and to generate a plane Orthophoto Image.

GF-1 series satellites use the RPC model to complete orthorectification. In this paper, the ENVI software is used to process the orthorectification of the GF-1 multispectral image.

(2)   Geometric correction: in this study, the geographic coordinates were corrected using the intersection of longitude and latitude network in the sea chart.

2.3.4. Water Depth Point Information

Combined with the sea chart and remote sensing image introduced in Section 2.2, a total of 596 water depth sample points were extracted. Because of the human activities such as fishery breeding, sea reclamation and so on, as well as the presence of clouds and flare areas in remote sensing images, the information of sample points located in such areas will differ greatly from the actual situation, which will become the abnormal points affecting the inversion effect of water depth.

To prevent these outliers from affecting the experimental results, we used the following strategies to screen out outliers:

Step 1: Use the classical log-linear model to perform a water depth inversion work in advance with all sample points;

Step 2: Calculate the absolute value of the difference between the predicted value and the observed value at each point, and calculate the standard deviation of predicted values.

Step 3: If the absolute value at a certain point is bigger than the standard deviation value in step 2, we can set this point as a suspected abnormal point. After this, we need to determine whether the point is located in the flare area or the breeding area through visual interpretation. If so, it must be screened out of the sample set.

After all the outliers were removed, 580 points are finally obtained, as shown in Figure 3.

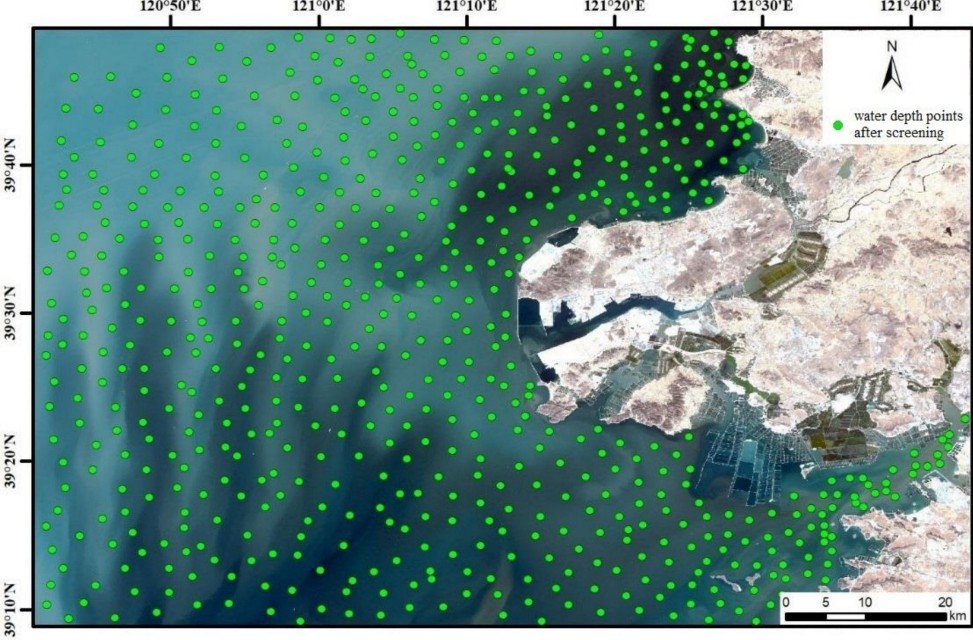

**Figure 3.** Distribution of water depth points, the green points in sea areas are the obtained samples after screening.

In this paper, with a predetermined split proportion, all 580 water depth points will be randomly divided into control points set and check points set. These two sets are commonly called "training data" and "test data" in the machine learning field. To prevent possible errors caused by a random split, each experiment will conduct multiple groups of repeated training based on different random splits and take the average values of evaluation indices as the final results.

## 3. Optical Remote Sensing Imaging Mechanism of Underwater Topography

### 3.1. Imaging Mechanism in the Clear Sea Area

Sunlight can see through the water body. Under the reflection of the shallow sea bottom sediment, the sunlight will pass through the upper seawater twice and be received by the optical remote sensor

after passing through the atmosphere. As shown in Figure 4, the signal received by the remote sensor contains $L_p$ (the light scattering information of atmosphere), $L_s$ (the reflection information of sea surface), $L_w$ (the light scattering information of waterbody) and $L_b$ (the reflection information of seafloor). Among them, the information $L_b$ that reflected from the seafloor into the optical remote sensor reflects the underwater topography, which is the direct physical basis of passive optical remote sensing inversion of shallow water depth.

$$L_t = L_p + L_s + L_w + L_b \tag{2}$$

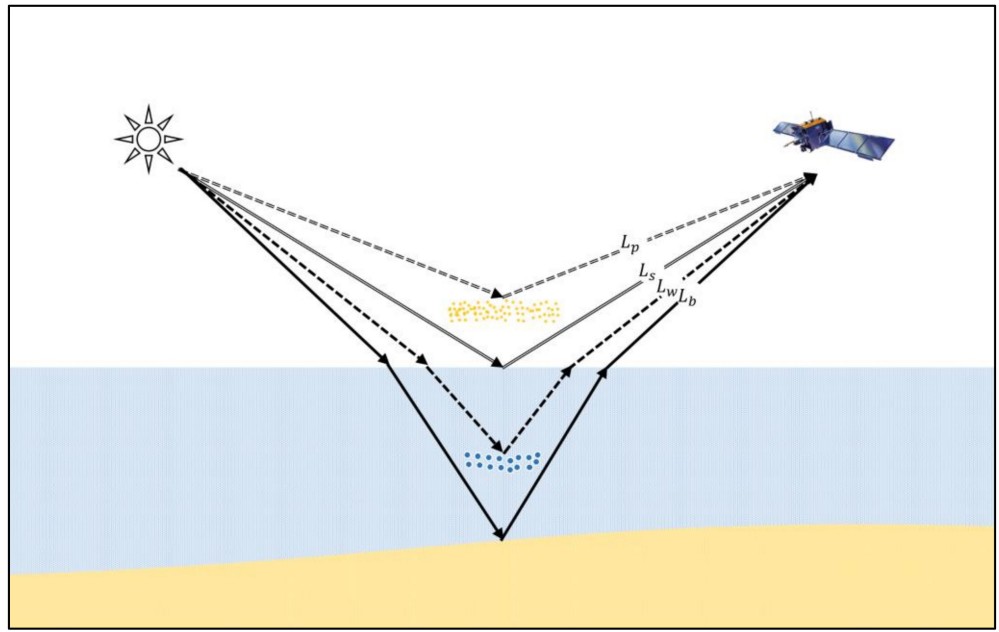

**Figure 4.** Imaging principle of underwater topography in the shallow or clear waters. The signal received by the sensor in this figure contains 4 parts: $L_p$ (the light scattering information of atmosphere), $L_s$ (the reflection information of sea surface), $L_w$ (the light scattering information of waterbody) and $L_b$ (the reflection information of seafloor).

In addition, the attenuation coefficient of light in the water body determines whether the light can reach the seabed or not, and it also determines whether the reflected light from the seabed can come out of the water surface. The attenuation coefficient determines the depth of the light in the water body that can be observed through perspective; this is why remote sensing imaging of underwater terrain based on light reflection information is limited to clear waters.

*3.2. Imaging Mechanism in the Turbid or Relatively Deep-Sea Area*

Changes in the submarine topography will modulate the flow of seawater under the sea surface. In areas where the seawater changes from deep to shallow, the velocity of the water will gradually increase, while in areas seawater changes from shallow to deep, the velocity will gradually slow down [39]. The regular changes of water flow will change the slope of micro-scale waves on different scales [30], forming the amplitude convergence and dispersion areas of micro-scale waves.

Under suitable observation angle, solar zenith angle, and other observation geometries, the difference of light reflection and scattering will be formed in the non-flare areas due to the change of micro-scale waves slopes in local areas; that is, the light intensity is alternately distributed in the micro-scale wave amplitude convergence and dispersion areas caused by terrain changes. The alternation of light and dark in optical remote sensing images indirectly represents the changes of underwater topography; the simulation process is shown in Figure 5. It is the accumulation of

reflections that determines the sea surface radiance in the amplitude convergence and divergence areas. The more reflection wave-fronts with suitable slope value, the stronger the radiance received by the remote sensor, and vice versa.

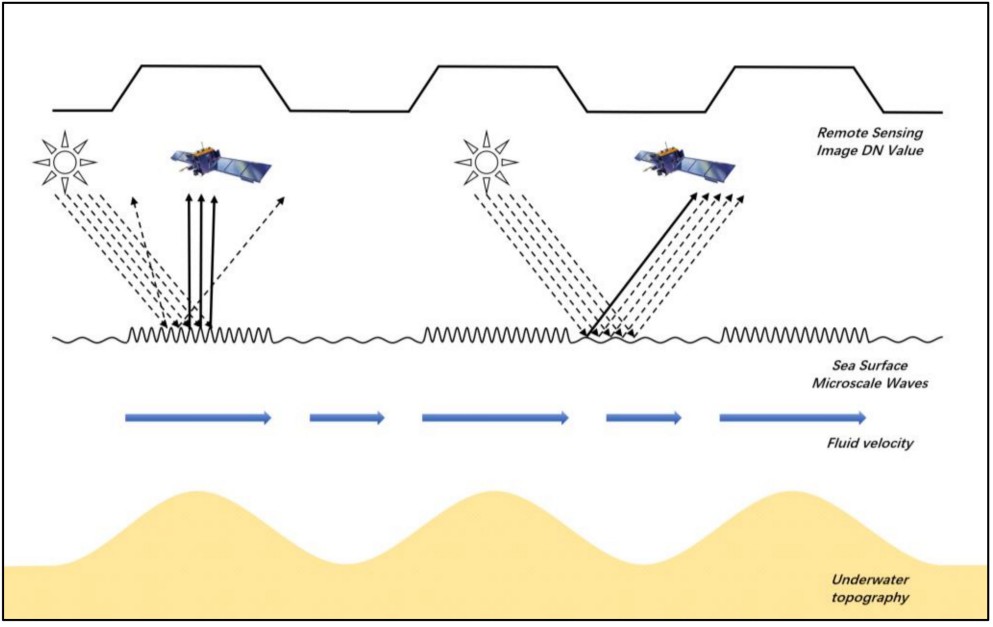

**Figure 5.** Imaging principle of underwater topography in the deep or turbid waters.

According to the comprehensive analysis of Zeisse [25], we obtain for the sun glitter radiance $N_{sg}$ received by the remote sensor with a view angle of no more than 80° is:

$$N_{sg} = H_\odot \frac{1}{4} p(z_x, z_y) [cos\beta]^{-4} [cos\theta]^{-1} \rho(\omega) \tag{3}$$

In the above formula, $H_\odot$ is the solar irradiance on the sea surface, $\beta$ is the inclination angle of the sea surface, $z_x$ and $z_y$ are the slope components that satisfy the specular reflection of sunlight to the remote sensor, $\omega$ is the reflection angle of the specularly reflected sunlight, and $\rho(\omega)$ is the Fresnel reflection Coefficient of the water surface. $p(z_x, z_y)$ is the probability distribution function of the slope, which can be approximately described by the following Gaussian equation [31]:

$$p(z_x, z_y) = \frac{1}{2\pi\sigma_x\sigma_y} exp\left[-\frac{1}{2}\left(\frac{z_x^2}{\sigma_x^2} + \frac{z_y^2}{\sigma_y^2}\right)\right] \tag{4}$$

$\sigma_x$ and $\sigma_y$ represent the surface roughness of the ocean in the directions of crosswind and upwind. It is found [31] that the sea surface roughness is a linear function of wind speed and shows a small degree of asymmetry with respect to the wind direction. The function relationship is shown in Equation (5), where $w$ is the wind speed in $m/s$.

$$\sigma_x^2 = 0.003 + 0.00192w \quad \sigma_y^2 = 0.000 + 0.00316w \tag{5}$$

In the work of passive optical remote sensing detection of sea water depth, we can supplement the inversion information of shallow water depth in the previous article based on the above conclusions. However, due to the diversity of submarine topography, water flow, and the complexity of the solar flare radiance model, to accurately describe the process mathematically, a very complex physical model must be established. To simplify this work, we chose to train the GRU neural network model by sampling water depth points, and use machine learning to construct the relationship between water depth and remote sensing image radiance.

## 4. Model and Algorithm

### 4.1. Analysis of the Unified Inversion Model of Water Depth in a Composite Environment

In the previous section, the imaging mechanisms of underwater topography optical remote sensing in two different underwater environments are given. Their models essentially give the mapping relationship between image spectral characteristics and water depth value. However, in the underwater optical remote sensing inversion of actual scenes, the regional seabed topography is often complex and changeable; shallow water and deep water situations coexist alternately. This is a typical composite underwater environment, which makes it difficult to uniformly characterize the underwater topographic inversion model of this kind of sea area using a certain mapping relationship. To solve the above problems, we propose to use the form of piecewise function to express the two mapping relations uniformly:

$$Z = \begin{cases} F_1(x, y, L), & (x, y) \in A \\ F_2(x, y, L), & (x, y) \in B \end{cases} \tag{6}$$

Among them, $Z$ is the water depth value of spatial location $(x, y)$, $L$ is the remote sensing image spectrum of this location, $A$ and $B$ separately represent the clear shallow water area and the deep turbid area, and $F_1$ and $F_2$ represent the mapping relationship between the spectral features of remote sensing image and the water depth to be measured in these two areas respectively.

However, in the practical work of water depth inversion, there is usually not enough information to define the spatial range of shallow waters and deep waters. That is, it is difficult to determine the definition domain of the piecewise function in Formula (5) in advance. The real underwater topography is usually similar to the form shown in Figure 6, which is only a simulation figure rather than a real environment. The clear shallow areas A and the deep turbid areas B tend to mix, and it is difficult to determine the boundary between A and B, which makes it difficult to apply the two imaging mechanisms in the previous section to different spatial regions. If there is an informatics tool that can approximate the piecewise function by numerical calculation, the trouble caused by the unclear boundary between areas A and areas B can be easily solved.

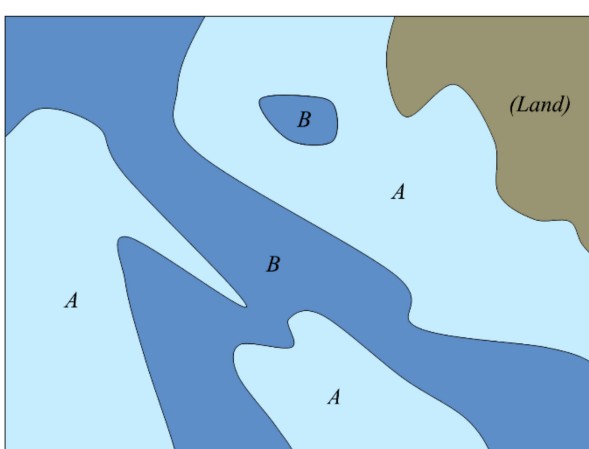

**Figure 6.** Real seabed topography, *A* represents the clear shallow water area, and *B* represents the deep turbid area.

### 4.2. GRU-Based Underwater Topography Optical Remote Sensing Inversion Algorithm

Deep learning has strong representation and generalization capabilities [40]. It can effectively count and summarize complex features and conditions in practical problems and usually has enough model capacity to fully fit the undetermined mapping relationship. Therefore, with the help of the excellent high-dimensional fitting ability of deep learning, a unified learning model is designed to approximate the water depth value in a composite ocean environment.

Unlike classical neural network models, the GRU model adopts a special neural unit structure. Similar to the LSTM model, the GRU model also uses a gate unit to store historical information and long-term status. The GRU unit has two gates, a reset gate and a update gate. The reset gate determines how the new input information is combined with the previous memory. The Update Gate defines the proportion of the previous memory saved to the current stage. The general form of the GRU unit is as follows:

$$g(x) = \sigma(Wx + b) \tag{7}$$

where $\sigma(x) = 1/(1 + exp(-x))$, which is the classical activation function sigmoid function in deep learning, $W$ and $b$ represent the weight matrix and bias vector of the network, which will gradually approach the optimal value during the training process of the model.

Figure 7 shows the internal structure of a GRU unit. At time $t$, the input of the GRU unit includes the hidden layer state variable of the previous time $h_{t-1}$ and the input information at the current time $x_t$. Then the model uses reset gate $r_t$ and update gate $z_t$ in turn to calculate the hidden layer state variable $h_t$. Finally, $h_t$ will be used as the unit's calculation result at time $t$, and passed to the next time for calculation.

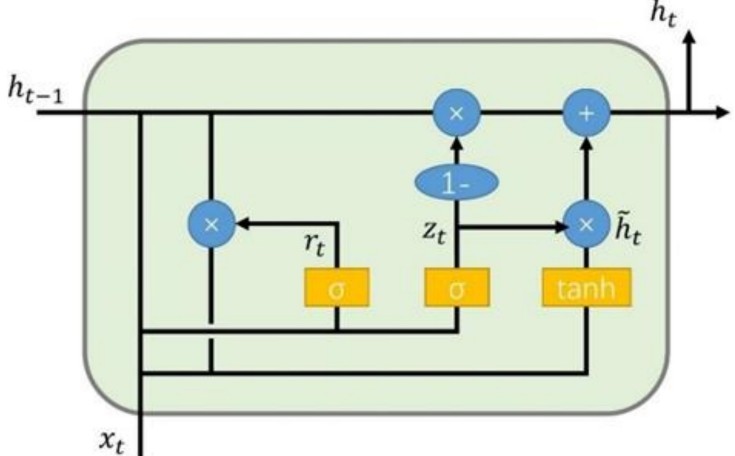

**Figure 7.** Structure of the gate recurrent unit (GRU) unit.

The GRU model can effectively learn the valuable features in spectral data. Compared with the previous semi-theoretical and semi-empirical models that are manually selecting key band combinations, the use of the GRU model greatly improves the feature extraction efficiency and inversion accuracy. Compared with the traditional neural network which regards the band information in the spectral dimension as discrete individuals, the GRU model, due to its recurrent network structure, can fully consider the information of a single feature and the overall feature sequence relationship with the spectral dimension. In addition, due to the existence of a long short-term memory mechanism, this kind of model can effectively solve the problems of gradient vanishing and gradient explosion problems in long-term training, fully retain the important spectral features appeared in the early stage, and fully learn the sequence relationship in spectral dimension features.

Based on the GRU model, this paper proposes a new unified water depth inversion framework and establishes a complex mapping relationship between the spectral characteristics of remote sensing image and the water depth of different depth sections for the complex sea areas where deep water and shallow water coexist. The model design is shown in Figure 8, including two GRU layers with a large number of GRU units, a fully connected layer, and a linear activation function for outputting predicted water depth values. To give full play to the excellent learning ability of the GRU model, we will invest all the band information to carry out the experiment, in order to obtain better results as far as possible.

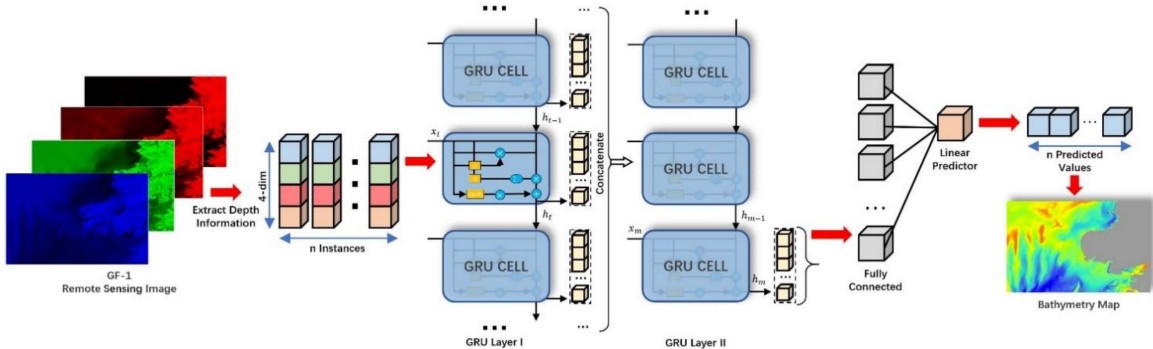

**Figure 8.** Diagram of the model.

In addition, some hyperparameters need to be determined in advance during the establishment and training of the GRU model, including the number of units in each GRU layer, optimizer selection, batch size, etc. The selection of these elements will have some impact on the final inversion accuracy of the model, and their selection will be discussed in detail in the next section.

## 5. Results and Discussion

### 5.1. Accuracy Evaluation Method

In this paper, root means square error (RMSE), mean absolute error (MAE), mean relative error (MRE) and determination coefficient ($R^2$) are used to evaluate the accuracy of water depth inversion and to analyze the water depth inversion effect under the inversion strategy. Among them, RMSE, MAE, and MRE are used to evaluate the error between the inversion results and the observed values. The smaller these values, the better the inversion effect. $R^2$ is also known as the fitting index, which, as its name implies, describes how well the inversion model fits the observed values. The range of $R^2$ is [0, 1], and the closer its value is to 1, the more consistent the inversion results are with the true distribution of observed values.

1.  RMSE (root mean square error)

$$RMSE = \sqrt{\frac{\sum_{i=1}^{n}\left(A_i - A_i^T\right)^2}{n}} \tag{8}$$

2.  MAE (mean absolute error)

$$MAE = \frac{\sum_{i=1}^{n}\left|A_i - A_i^T\right|}{n} \tag{9}$$

3.  MRE (mean relative error)

$$MRE = \frac{\sum_{i=1}^{n}\left(\left|A_i - A_i^T\right|/A_i^T\right)}{n} \tag{10}$$

In the above three formulas, $A_i$ and $A_i^T$ are the inverted water depth value and the true water depth value of the $i$th point, respectively, and $n$ is the total number of water depth points participating in the accuracy evaluation.

4.  $R^2$ (determination coefficient)

$$R^2 = \frac{SSR}{SST} = 1 - \frac{SSE}{SST} \tag{11}$$

In the above formula, *SST* is the total sum of squares, *SSR* is the regression sum of squares, and *SSE* is the error sum of squares.

*5.2. Overall Accuracy Evaluation of Underwater Topography*

First, from the perspective of the overall water depth, the four accuracy evaluation methods given in Section 5.1 are applied to evaluate the inversion effect of the four water depth inversion models (log-linear model [7], Stumpf model [9], MLP model, GRU model [33]), root mean square error (RMSE), mean absolute error (MAE), mean relative error (MRE) and determination coefficient ($R^2$) are obtained, respectively.

As can be seen from Table 2, GRU models have significantly improved accuracy when compared with semi-theoretical and semi-empirical models such as the four-band log-linear model and the Stumpf model, as well as MLP statistical models. The RMSE of the GRU model's inversion results is 3.69 m, the MAE is 2.72 m, and the MRE is 19.6%, indicating a positive inversion effect. For the other three models, the Stumpf model performs the most unsatisfactory in overall inversion, with RMSE of 10.2 m, MAE of 8.1 m, and MRE of 91.6%. Since the four-band log-linear model contains more band information, a slightly better result was achieved in the complex underwater environment of the study area, with three indices of 6.9 m, 5.2 m, and 50.4%, respectively. For the MLP statistical model, the three indices are 6.3 m, 4.8 m, and 30.4%, respectively. It can be seen that compared with the semi-theoretical and semi-empirical model, the statistical model can better deal with problems in complex and diverse environments.

**Table 2.** The overall accuracy evaluation.

| Model | $R^2$ | RMSE (m) | MAE (m) | MRE (%) |
|---|---|---|---|---|
| Log linear | 0.60 | 6.90 | 5.20 | 50.4 |
| Stumpf | 0.16 | 10.20 | 8.10 | 91.6 |
| MLP | 0.69 | 6.30 | 4.80 | 30.4 |
| GRU | 0.88 | 3.69 | 2.72 | 19.6 |

Figure 9 shows the inversion scatter diagram of the four models at the checkpoints, which can more directly judge the inversion effect of these four models. The higher the determination coefficient $R^2$ is, the more water depth points converge to the standard measurement line, and the better the model fitting is. The lower the determination coefficient, the more divergent the water depth points are to the trend line. As can be seen from the figure, the GRU model has a higher degree of regression than the first three models, with a determination coefficient $R^2$ of 0.88, while the Stumpf model is 0.16, the four-band log-linear model is 0.60, and the MLP model is 0.69, all of which are significantly lower than the GRU model. It can also be seen from the scatter distribution that the effort of the four-band log-linear model and MLP model in deep water area is relatively poor, and it is difficult to accurately describe the water depth information of areas deeper than 25 m. The Stumpf model is even less satisfactory and can only accurately describe the depth information in the middle depth areas. The GRU model has a concentrated scatter distribution, and the trend line is approximate to the line $y = x$, thus obtaining an ideal inversion result.

*5.3. Segmented Accuracy Evaluation of Underwater Topography*

In the precision analysis of the water division deep section, the errors will be calculated piecewise according to the prediction results of the four models in Section 5.2. According to the measured water depth, the checkpoints were divided into four groups of segmented point sets, including 0–8 m, 8–16 m, 16–24 m, and 24–32 m, and the MAE and MRE of each model were obtained, respectively. The calculation results are shown in Figure 10.

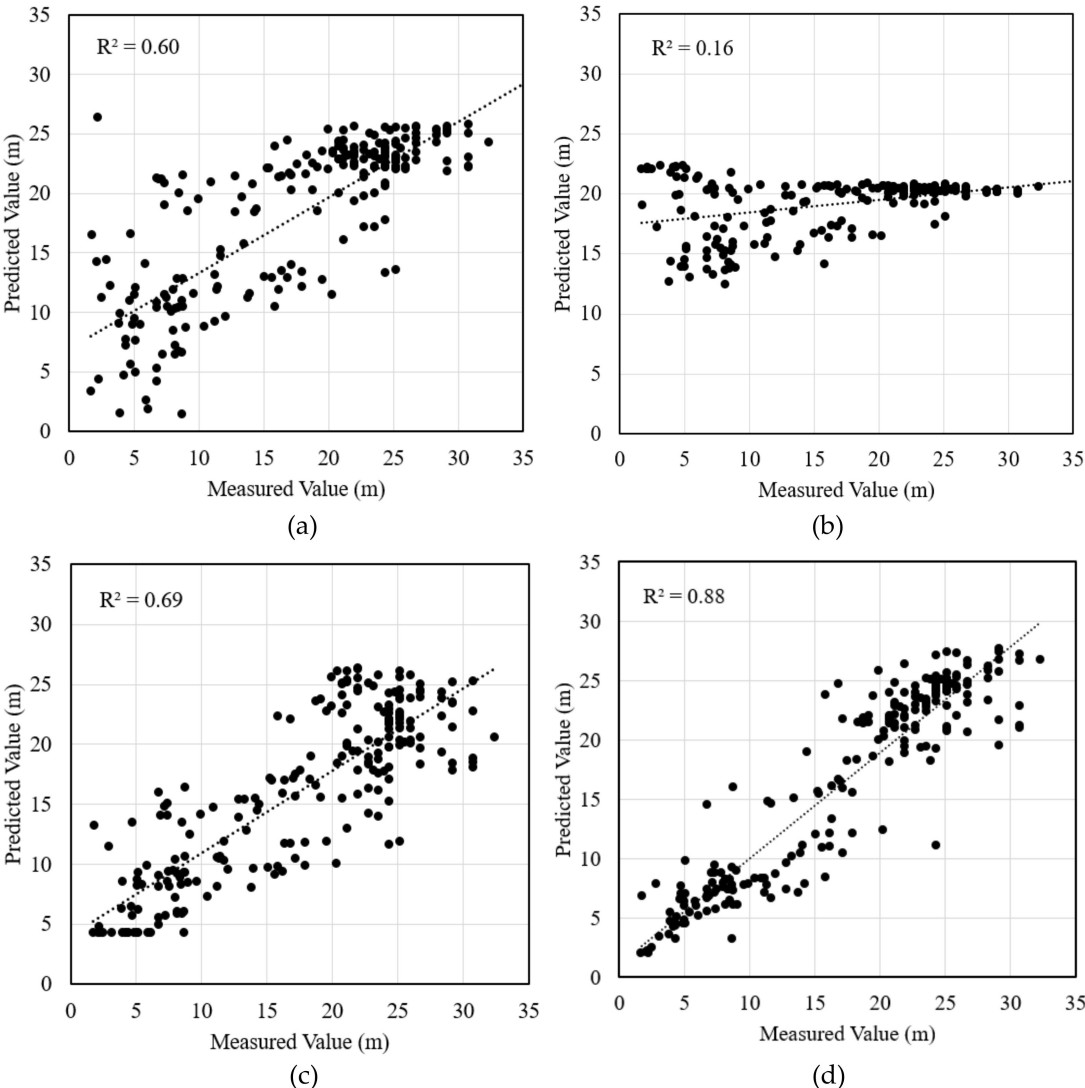

**Figure 9.** Comparison of the inverted depths and the observed depths: (**a**) the scatter plot of the log-linear model results; (**b**) the scatter plot of the Stumpf model results; (**c**) the scatter plot of the MLP model results; (**d**) the scatter plot of the GRU model results.

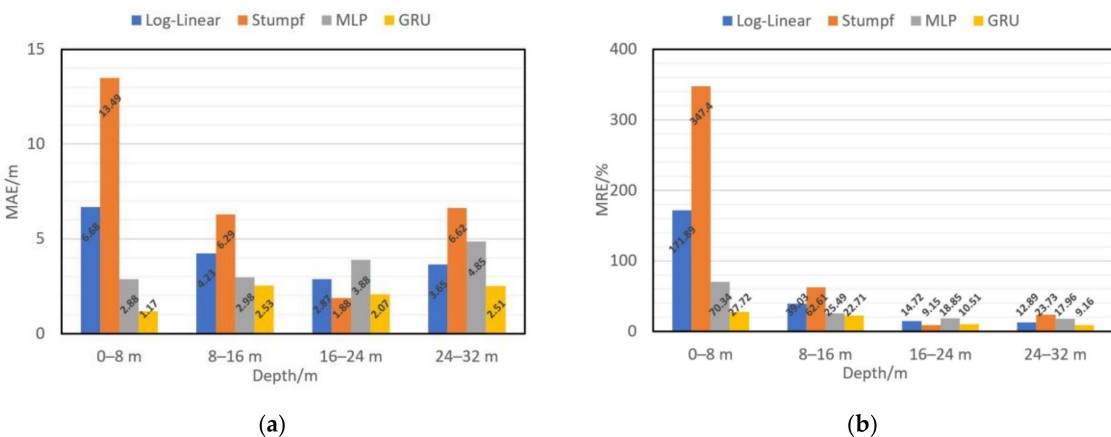

**Figure 10.** Inversion error statistics of different models at different water depth segments: (**a**) mean absolute error (MAE) of different models at different depth segments; (**b**) mean relative error (MRE) of different models at different depth segments.

It can be seen from the figure that, compared with the semi-theoretical and semi-empirical models and the traditional statistical model, the GRU model has achieved considerable advantages in almost all water depth segments. Only in the depth range of 16–24 m, the Stumpf model gained a small advantage. The performance advantage of the GRU model is especially obvious in the deep (24–32 m) and shallow (0–8 m) areas. In the deep-water area (24–32 m), due to the lack of optical information in the deep area and the small number of water depth samples in this area, the performance of each model is reduced to a certain extent compared with the shallower research areas. However, the GRU still achieves relatively low segmented inversion errors. In the shallow water area (0–8 m), due to the limited global learning ability of other models, especially classical methods, poor inversion results are often obtained. However, the GRU model can still maintain high inversion accuracy in this area, which shows that the GRU model is indeed suitable for the unified inversion work in deep and shallow waters.

### 5.4. Influence Analysis of Model Parameters

For deep learning methods, the adjustment of the learning rate, batch size, network structure and other hyperparameters usually has a huge influence on the final effect of the model. In this part of the paper, we will carry out comparative experiments from various perspectives and strive to obtain a set of hyperparameter combinations with better effects to provide a model basis for the subsequent overall inversion results. To ensure the fairness and rationality of the experiment, when the comparison experiment is carried out for a certain hyperparameter, the default value of the model or the recommended value of the model will be uniformly used for other hyperparameters.

#### 5.4.1. Network Structure

The differences between different network structures mainly lie in the number of hidden layers and the number of neuron nodes in each layer. This section will carry out comparative experiments and discussions on these two aspects. According to the scale of the problem, 1 to 3 hidden layers are designed for the model, and the specific node number of each hidden layer is shown in Table 3. The comparison index covers the four accuracy evaluation indices in Section 5.1 and the training time of the model.

**Table 3.** Error statistics of different network structures.

| Structure | $R^2$ | RMSE (m) | MAE (m) | MRE (%) | Time (s) |
|---|---|---|---|---|---|
| 50 | 0.83 | 4.53 | 3.20 | 20.71 | 191 |
| 100 | 0.82 | 4.62 | 3.20 | 20.52 | 200 |
| 200 | 0.80 | 4.84 | 3.40 | 21.73 | 217 |
| 400 | 0.81 | 4.74 | 3.40 | 20.90 | 298 |
| 25–50 | 0.85 | 4.24 | 3.01 | 19.08 | 322 |
| 50–100 | 0.92 | 3.13 | 2.26 | 14.96 | 316 |
| 100–200 | 0.92 | 3.13 | 2.25 | 15.01 | 345 |
| 200–400 | 0.91 | 3.26 | 2.29 | 14.78 | 515 |
| 400–800 | 0.91 | 3.34 | 2.39 | 14.85 | 1059 |
| 25–50–25 | 0.83 | 4.47 | 3.11 | 18.55 | 482 |
| 50–100–50 | 0.84 | 4.41 | 2.96 | 17.75 | 446 |
| 100–200–100 | 0.90 | 3.38 | 2.38 | 15.56 | 467 |
| 200–400–200 | 0.91 | 3.24 | 2.28 | 14.98 | 680 |

As can be seen from Table 3, compared with other alternative network structures, the model of double hidden layer structure with 100 and 200 nodes has achieved the best value in multiple evaluation indices, and the model has achieved a good balance between performance and efficiency. Therefore, the network structure will be preferred in the follow-up experiments.

5.4.2. Optimizer Selection

The main work of neural network training is to update parameters and optimize the objective function, so the selection of the optimizer is also an important work affecting the model effect. Common optimizers include SGD, RMSprop, Adagrad, Adadelta, Adam, Adamax, etc. We will carry out comparative experiments with the above six optimizers. The indices of the experimental results are shown in Table 4, and the MAE variation trends of each model on the validation set are shown in Figure 11.

**Table 4.** Error statistics of different optimizers.

| Optimizer | $R^2$ | RMSE (m) | MAE (m) | MRE (%) |
|-----------|-------|----------|---------|---------|
| SGD | 0.85 | 4.26 | 3 | 22.06 |
| RMSprop | 0.91 | 3.21 | 2.32 | 16.89 |
| Adagrad | 0.8 | 4.89 | 3.47 | 23.69 |
| Adadelta | 0.91 | 3.21 | 2.22 | 15.72 |
| Adam | 0.92 | 3.13 | 2.19 | 15.39 |
| Adamax | 0.91 | 3.18 | 2.27 | 16.35 |

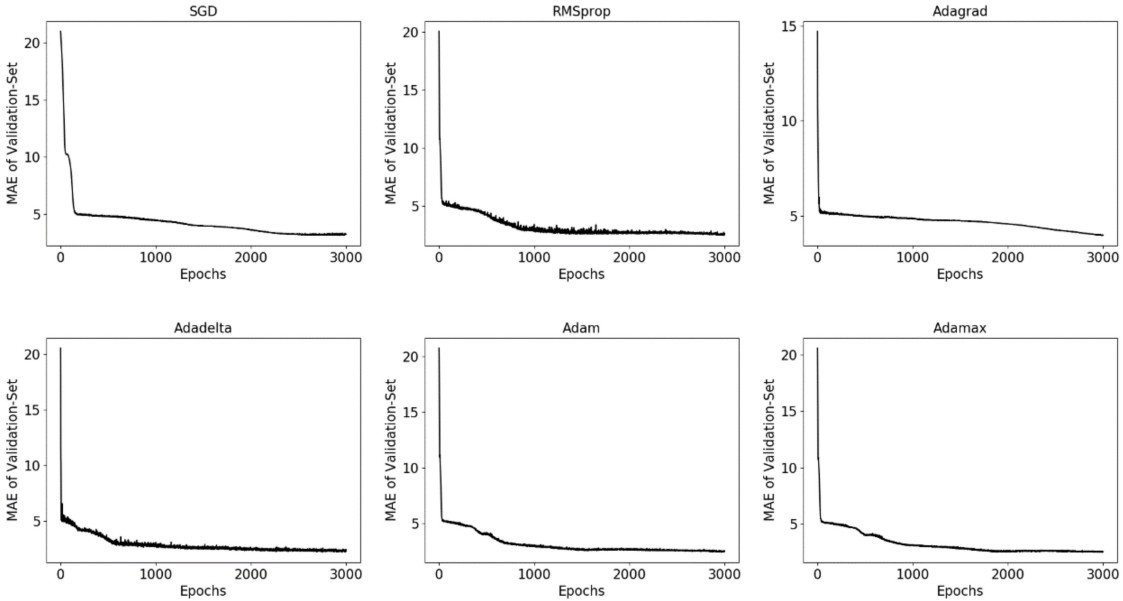

**Figure 11.** Change trend of MAE.

It can be seen that after a certain number of iterations, the optimizer Adam achieves the optimal inversion effect by considering the regression accuracy of the model and the stability of the training process. Therefore, this optimizer will be preferred in the following experiments.

5.4.3. Batch Size

Batch-size is the number of samples sent into the model during each round of neural network training. A larger batch-size can usually make the network converge faster, but too large a batch-size will consume many memory resources and require more iterations to meet the model training, so we need to select a suitable size of batch-size for training. We respectively select different numbers as batch-size to carry out the comparison experiment, and the indices of the experimental results are shown in Table 5.

**Table 5.** Error statistics of different batch size.

| Batch Size | $R^2$ | RMSE (m) | MAE (m) | MRE (%) | Time (s) |
|------------|-------|----------|---------|---------|----------|
| 8 | 0.91 | 3.3 | 2.38 | 15.86 | 1468 |
| 16 | 0.91 | 3.25 | 2.34 | 15.27 | 727 |
| 32 | 0.91 | 3.28 | 2.39 | 15.74 | 393 |
| 64 | 0.91 | 3.21 | 2.31 | 15.11 | 202 |
| 128 | 0.89 | 3.54 | 2.62 | 17.14 | 108 |
| 256 | 0.88 | 3.72 | 2.76 | 18.73 | 70 |

As can be seen from the above table that in this depth inversion work, when batch-size is set to 64, the model achieves the best efficiency and performance. Therefore, in the follow-up experiments, batch-size will be preferred to take this value.

### 5.4.4. Number of Iterations

The number of iterations refers to the number of times that the entire training set is input into the neural network for training. Usually, sufficient iterations are required to enable the model to fully learn the information in the training data and to fully build the deep learning model. However, this does not mean that the more iterations, the better. Too many iterations will cause the model to overlearn the information of training data, resulting in the phenomenon of "overfitting" and leads to an increase of the error on the test set.

For this reason, we carried out comparison experiments with different iteration times, and the model structure we use is the 100–200 double hidden layer network structure recommended in Section 5.4.1. The inversion accuracy of the model at control points and check points are calculated, respectively, and the results are shown in Figure 12. It can be seen that when the number of iterations is between 2000 and 2500, the MAE curves of the training set and the test set intersect once. When the number of iterations reaches about 2500, the MRE of the training set and the test set is the same. Continuous training will further reduce the MAE and MRE of the training set, but will not reduce the errors of the test set. The model was overfitted to the training data at this time. Therefore, for the model of 100–200 double hidden layer network structure, it is a good choice to choose the number of iterations between 2000 and 2500.

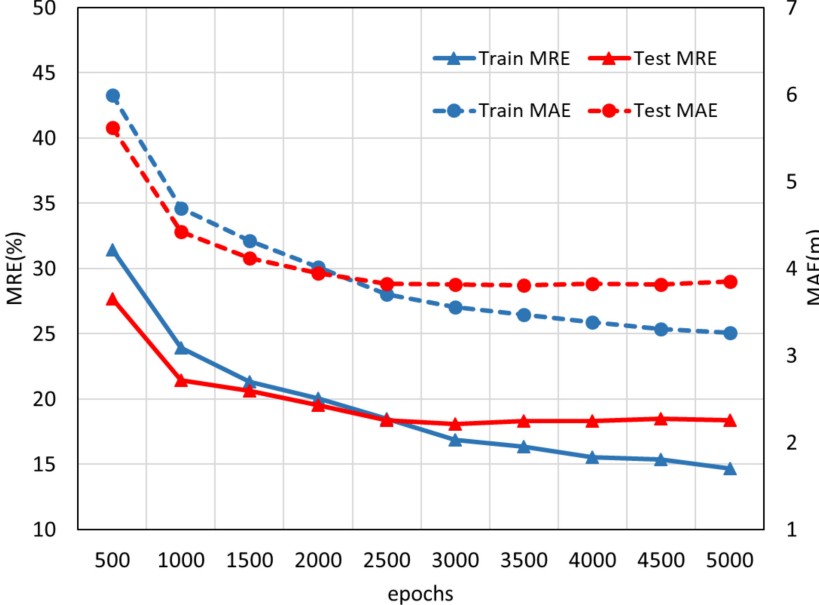

**Figure 12.** Change trend of MRE and MAE.

*5.5. Influence of Control Points Proportion*

For common deep learning problems, to train a model as complete as possible, we usually want the training set to be as rich as possible, including massive training data and high data dimensions. However, for the water depth optical inversion problem, it is difficult to obtain large training data due to the objective difficulties in acquiring the depth control points, which poses a challenge to the usability of the model under the condition of limited samples.

To this end, we carried out experiments by adjusting the proportion of water depth control points. On the premise of maintaining the same other conditions, the proportion of the control points is taken as 10%, 20%, 40%, 60%, 80% to experiment, respectively, and the experimental results are shown in Table 6. It can be seen that for the problem in this paper with 580 water depth points, as the scale of the control points gradually increases, the evaluation indices are all gradually improved, and this improvement process is especially obvious when the proportion of the control points does not exceed 40%. When the proportion 40%, the determination coefficient $R^2$ is increased to 0.89, MRE is 20.33%, RMSE and MAE are 3.78 m and 2.78 m. However, if the size of the control points set is further improved, the improvement of evaluation indices is relatively limited. It can be seen that 40% control points (232 points) can meet the needs of water depth optical inversion in this area.

**Table 6.** Error statistics of different trainset size.

| Control Point | $R^2$ | RMSE (m) | MAE (m) | MRE (%) | Time (s) |
|---|---|---|---|---|---|
| 10% | 0.82 | 4.75 | 3.35 | 27.04 | 103 |
| 20% | 0.85 | 4.37 | 3.20 | 24.11 | 149 |
| 40% | 0.89 | 3.78 | 2.78 | 20.33 | 219 |
| 60% | 0.90 | 3.49 | 2.50 | 19.55 | 391 |
| 80% | 0.92 | 3.24 | 2.33 | 18.20 | 508 |

*5.6. Spatial Analysis of Underwater Topography Inversion by Remote Sensing*

Based on the water depth points mentioned in Section 2.3.4, a GRU neural network model is established and trained. Moreover, the water depth inversion process of the research area mentioned in Section 2.2 is carried out. Figure 13 is the inversion map of water depth in the study area of the Liaodong shoal, and the blank part on the right is the land part processed by masking.

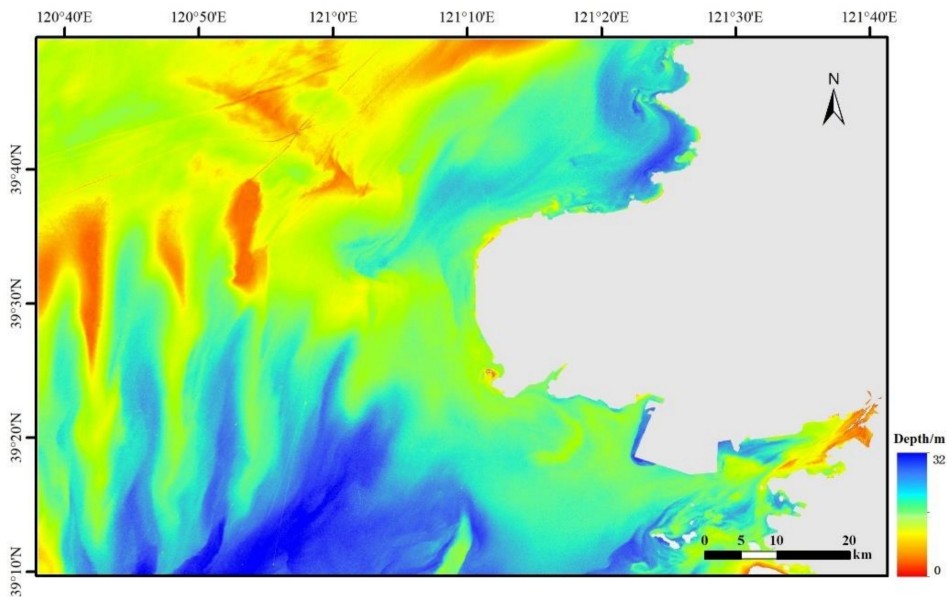

**Figure 13.** Water depth inversion results.

From the water depth inversion results, it is obvious that there are several radial tidal ridges in the northern part of the area. With regular distribution and frequent changes in water depth, most of the water depth values are between 15 m and 32 m. The southern area, namely, the Laotieshan Waterway, has a dramatic change in topography. The deepest area is over 30 m-deep, while the shallowest place is only within 10 m-deep. There is obvious relief in this area, and the slope is also large. The local slope can reach 1–4‰. The results of this bathymetric inversion are consistent with the actual seabed topography characteristics in this research area, which also verifies the correctness of this experiment once again.

### 5.7. Considerations about the Input Data for Model

The application of the unified water depth inversion framework and GRU model in this paper provides a new perspective and method for the passive optical remote sensing water depth inversion. However, there are still some issues that need further discussion in this study:

(a) GF-1 satellite is a multi-spectral satellite with a spatial resolution of 16 m and has four optical bands, which cannot make full use of the sequential feature learning ability of the GRU model. If reliable hyperspectral data are introduced to future work, it is believed that better results will be obtained.

(b) The Liaodong Shoal area has a relatively high sediment concentration; the sea water is muddy. Considering that some information such as chlorophyll concentration, yellow substance concentration and suspended substance concentration have considerable influence on the bathymetric optical signal, if these factors can be introduced into the future work, it is believed that the inversion performance of this method can be further improved. At the same time, the dimension of input data will be higher and more suitable for deep learning models.

(c) The effects of passive optical remote sensing depth inversion are often limited by the quality of control points and check points. Generally, the sources of bathymetric data include sonar measured data and scanning charts, and their accuracy is usually quite different. This makes the source of training sample points, the precision of data acquisition equipment and acquisition process, and the spatial distribution of sample points all worthy of further study.

(d) Deep learning is a data-driven model method. To ensure the learning effect of the deep model, abundant and diverse training data are required in the training process. However, in the water depth inversion work, the lack of sufficient samples is very common. "few shot learning" and "transfer learning" may provide solutions and research directions for this problem.

(e) The water depth control points are the basis of building the model, and their quality is very important. In the real scene, the image pixels corresponding to the water depth points in the sea chart are affected by the surface flare and the boundary of the aquaculture area, resulting in the distortion of the pixel spectrum of the remote sensing image, and the predicted values of these pixels are greatly different from the measured values. In fact, these pixels have been "polluted" and are no longer suitable as control points. It should be pointed out that the GRU model also pays attention to the number of control points, and maintaining a certain number of control points is the premise of training an effective model. In this paper, a total of 596 control points was collected. For the points with more than one standard deviation, we carried out a visual interpretation to ensure that they were "polluted" pixels, and 16 points were deleted, and 97% of the points were retained. Therefore, on the premise of ensuring the quality of water depth control points, the quantity of control points is also guaranteed.

## 6. Conclusions

Based on GF-1 satellite remote sensing data, this paper proposed a unified remote sensing inversion framework for water depth in a composite environment, applied and adjusted a GRU deep

learning model to carry out a water depth inversion experiment in the Liaodong Shoal. The main work and conclusions are as follows:

Based on the traditional passive optical inversion principle of shallow water depth, combining with the relationship between underwater topography, flowing water under the sea surface, distribution of the sea surface micro-scale wave, and local flare brightness in non-flare area, this paper analyses the mechanism of passive optical remote sensing inversion in turbid or deep-sea area, and proposes a new inversion framework for water depth, which can simultaneously satisfy the requirements of optical remote sensing depth inversion for shallow and deep areas.

The overall and segmented water depth inversion effects were evaluated, respectively. Through the comparative analysis of various indices, it is found that the GRU model has achieved leading results compared with other traditional inversion methods, both in terms of the overall inversion effect and the local inversion performance in each depth segment. Around the research area of Liaodong Shoal with a complex environment, the model optimization and result analysis are carried out in many aspects. For this research area, the (100–200) double-hidden layer network structure is applied, the Adam optimizer is called to optimize the model, the batch-size is determined to be 64, the training iterations from 2000 to 2500 are used, and determine that at least 40% of sample points can meet the needs of model training.

**Author Contributions:** Conceptualization, Z.L. and J.Z. (Jie Zhang); methodology, Z.L. and Y.M.; software, Z.L.; validation, Z.L., Y.M., J.Z. (Jie Zhang) and J.Z. (Jingyu Zhang); formal analysis, Y.M. and J.Z. (Jie Zhang); investigation, J.Z. (Jingyu Zhang); resources, J.Z. (Jingyu Zhang); data curation, Y.M.; writing—original draft preparation, Z.L.; writing—review and editing, Z.L., Y.M. and J.Z. (Jingyu Zhang); visualization, Z.L.; supervision, Y.M.; project administration, J.Z. (Jie Zhang); funding acquisition, Y.M. All authors have read and agreed to the published version of the manuscript.

**Funding:** This research was funded by the National Natural Science Foundation of China under contract Nos. 51839002 and 41906158.

**Acknowledgments:** The authors would like to thanks China's High-Resolution Earth Observation System Major Project for its support, as well as the GF-1 data provided by China Center for Resources Satellite Data and Application.

**Conflicts of Interest:** The authors declare no conflict of interest.

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
