# Peer review of "Underwater Topography Inversion in Liaodong Shoal Based on GRU Deep Learning Model"

_remotesensing, doi:10.3390/rs12244068_

Round 1
Reviewer 1 Report
Underwater Topography Inversion in Liaodong Shoal Based on GRU Deep Learning Model
The paper presents a deep learning based inverse bathymetry using data from hyperspectral satellite imagery. A GRU deep learning model is presented that uses estimates of depth from each imagery band and every pixel point to create a two-dimensional map of depths. The model is unique in the use of both shallow and deep-water algorithms for making initial estimates of the depth due to the different physical constraints for each water depth. The paper details the hyperparameter evaluation for the GRU model. Initial results are promising when compared to alternative inversion methods and the qualitative bathymetry estimate at Liaodong Shoal looks very good.
The reviewer thanks the authors for this paper and while the results are interesting, there are some questions that seem to be unanswered in the text. Additionally, the writing in the abstract and introduction could use to be edited for grammar.
- The GRU model is an interesting choice for this problem. The model is unique in the ability to hold onto history, but in the example shown in this paper there is no time history. All the reviewer can assume is the model is holding onto some spatial history. This opens the door for a few questions: Is the location of each pixel being passed to the GRU? If so, how does this differ from a fancy interpolation scheme? If not, does the order with which the points are given to the GRU matter?
- One of the unique features proposed in this paper is the ability to handle the depth inversion for both deep and shallow water points. The difficulties are described reasonably well in the text. How does the algorithm decide when to use deep vs shallow water algorithms for estimating depths? Based on Figure 8, the initial estimates of depth are performed outside of the GRU, so there must be a mechanism to determine what algorithm to use. Is a predefined mask, as shown in Figure 6 used to determine the location of shallow and deep-water methods? If so, how sensitive is this method when making an initial estimate of deep vs shallow water regions?
Additional comments are more directed to the language in the text, so the page and line numbers will correspond to the comments.
Introduction: The introduction could use a paragraph detailing the use of machine learning for depth inversion? This would help the reader better understand the gap in the present methodology for the estimate of depth as well as the present state of machine learning as a method for solving this difficult problem.
Page 2, line 63: The sentence is confusing and could use to be rewritten
Page 2, line 66: Please remove the sentence or provide a reference
Page 2, line 73: Please provide a reference about the difficulty of light attenuation and bottom reflection
Page 2, line 86: Could be useful to provide a sentence explanation as to why the deep-water methods don’t work in shallow water.
Page 2, line 90: please remove “an important model in the field of deep learning”. This is an opinion.
Page 3, line 98: please provide references for the uses of GRU
Page 4, line 151: please provide a reference for the FLAASH method
Page 5, line 168: In this line the authors describe throwing out points where a traditional inverse method differs from the true value by greater than a standard deviation. The reviewer feels the authors are biasing the prediction to the better by doing this. Shouldn’t the goal of the method be to improve on all estimates of depth inversion? Please provide a defensible reason for throwing out points.
Page 5, line 176: Just to be clear, are all points assumed to be independent? Do you provide any relationship between the points to the GRU model? Is the location of the points being passed into the GRU model?
Page 5, line 176-178: During the random splitting of points, how does the author determine the number of shallow vs deep points into the training set. How does the random sampling avoid collecting all points from a particular area if the points are assumed to be independent (the spatial location is ignored)?
Page 6, line 199: Please provide a reference for the velocity information.
Page 7, line 223: The assumption that surface roughness is independent of the wind direction seems like it would inject a lot of uncertainty into this method. Can you address why this is a reasonable assumption. In general, the reviewer would assume the micro scale surface roughness is influenced by the magnitude and direction of the winds.
Page 9, line 278-280: The reviewer is confused in this sentence about the usefulness of the GRU model. More clarity to the uniqueness of this model is needed? The description in this sentence is needs more
Page 10, line 323: Please remove one “Firstly”
Page 10, line 347: Between Table 2, the text on line 347, and Figure 9, the GRU model has different values of r2. Please provide a consistent value for the statistics or make sure it is clear why they are different.
Page 11, section 5.3: This section is confusing to the reviewer. Was the GRU model trained on these smaller subsets of data, or was the model trained as previously described and then tested only on subsets of data? If the model was only trained and tested on these subsets of depths, it seems the model would overfit to those depths. Please provide an explanation for the value of this section.
Page 12-15: the hyperparameter evaluation is well described, but the page numbers restart on page 13.
Page 15: lines 462-464: Please rewrite this sentence with more clarity.
Page 15: lines 471-472: The qualitative plot looks great, but the reviewer would prefer to see a quantitative 2D map. Generally, on depth inversion results, the author provides a side by side of true, estimate, and difference, so the reader can see the accuracy of the method.
Reviewer 2 Report
The authors present a new approach for satellite bathymetry retrieval using a deep neural network. This is an interesting study with potentially useful results which can have various applications. However the manuscript seems in draft mode. This manuscript needs major review before it can be considered for publication. The authors are encouraged to dedicate more time to assess all the points below in detail and provide an improved paper with high quality.
The manuscript has significant shortages in the following points:
- The current status of the topic (i.e.: satellite-derived bathymetry) is not sufficiently presented. The authors must cite and describe the most important studies so far.
- The objective of the study is not clearly defined. The authors must justify why their application with the GRU algorithm is important and they must provide enough references with similar applications in the field of artificial neural networks.
- The satellite sensor metadata (e.g.: angle with the sun) are not described in detail although the authors consider important the acquisition geometry. These should be included.
- The pre-processing steps (paragraph 2.3) of satellite data are not sufficiently explained. More information is required about all the corrections that were performed (e.g.: atmospheric, orthorectification, geometric). The radiance conversion must be sufficiently described and the equation parameters are not fully explained (how the gain and bias factors are calculated?).
- There is no explanation about the source of the bathymetry points used for training the algorithm. How these points were collected? Are they from sonar surveys or are they digitized from nautical charts? When these bathymetry points were collected and how reliable are these bathymetry points? The authors must explain in detail the filtering of outliers in the bathymetry points.
- Paragraph 3.2: There are not enough citations of previous studies on this topic. Description of the method is poor. Please provide more information and citations.
- Line223: the authors assume for the model that the surface roughness is independent of the wind direction. This is a condition that rarely occurs in situ and it should rather cause difficulties when applied in real-world data. What is the influence of this assumption on the resulting bathymetry (Figure 13)? Please discuss this issue in detail.
- Figures:
Figure 3: the map does not contain a legend and a scalebar. The caption needs to be improved (is too simple).
Figure 4: The caption is extremely poor. The authors must describe in detail all the components shown on the figure.
Figure 6 is not cited within the text. In addition, it is not mentioned how the areas A and B were calculated. This must be described properly.
- In the Discussion section, the authors must include a paragraph about the limitations of the method due to a) the imagery specifications (i.e.: band number, resolution) and b) the optical properties of the water column.
Reviewer 3 Report
The manuscript by Leng et al. presents a new method to retrieve the bathymetry for shallow and relatively deep waters based on the GRU model. The methodology is interesting and generally well described.
Also, while the text is generally grammatically correct, some language is awkward owing to word choice or overly complex sentences; a read through by a professional editor is necessary.
Other points are listed below by line number:
line 15: change "to obtain ideal" to "to provide optimal"
line 16 to 19: The sentence is tool long and hard to understand. Please rephrase.
line 20: Replace "And" at the beginning of the sentence with a better suited word.
line 24: -Replace "R2" by "R2"
-the determination coefficient R2 is higher
-the MRE is lower
line 28 and 32: replace "effect" with "performance"
line 32: replace "relatively ideal" with a better suited term.
line 36: replace "of marine science, which is of great significance to" with "for"
line 37: add "," after "construction"
line 39: replace "range" with "spatial coverage"
line 40: for the inversion of shallow water bathymetry
line 40: remove "and the sea area"
line 45: replace "developed" with "developing"
line 46: "mainly formed three forms" with "mainly in three different approaches"
line 51: "in physical" not clear
line 62 to 64: unclear sentence. Please rephrase ir for more clarity.
line 67: replace "expresse" with "express"
line 69: End the sentence after "reef detection". Start the next sentence with "However," rather than but.
line 73: replace "on" with "to"
line 73 to 75: please rephrase and make shorter sentences
line 75 to 78: remote repeated sentence "Previous studies have confirmed that the trend of seabed topography will have a regular impact on the water flowing under the sea surface,"
line 80 to 81: unclear sentence
line 97 to 98: please add references
line 112 to 115: please rephrase and make shorter sentences
line 115 to 117: please rephrase
line 120: in the south
line 127: replace ""Data Intorduction" with "Datasets"
line 133 to 135: replace "This paper uses the four-band data of multispectral remote sensing images, shown in Figure 2. 133 (a), and the imaging time is 03:11:58 (UTC) on April 8, 2016, the spatial resolution is 16m, the 134 interception size is 4300 × 5500, and the coordinate system is the WGS-84 Coordinate System." with "This paper uses the four-band data of the GF-1 multispectral image (Figure 2.a) referenced to the WGS-84 coordinate system, acquired on April 8, 2016 at 03:11:58 (UTC) with a spatial resolution of 16m, and a size of 4300 × 5500 pixels."
section 2.3: please add reference where required
line 155: remove "of"
line 158: replace "in this paper, with the help of the intersection of longitude and latitude 158 network in the sea chart, the geographic coordinates are corrected." with "in this study, the geographic coordinates were corrected using the intersection of longitude and latitude network in the sea chart."
line 168: Do not start a new paragpraph here.
line 174 to 176: please rephrase
line 194: replace "areas with clear area" with "clear waters" (in Fgiure 4 caption too)
line 208: remove "And"
line 214: add reference after "According to" or remove it.
line 230: replace "choose" with "chose"
line 235-236: please rephrase
line 255 to 257: please rephrase
section 5.1: No need for this section. These statistical metrics are widely known. They are mentioned in section 5.2.
line 323 to 327: please rephrase and divide into two sentences.
line 331: replace "ideal" with a better suited word
line 333: "Due to the fact"?
line 328 to 338: no need to repeat the values. They can be found in the table.
line 343: replace "At the end of" with "The higher"
line 348: replace "worse" with "lower"
line 347: Why the R2 of the GRU model in the text is different than the R2 in Figure 9?
line 367: "remove" research
Figure 10: Same information as Table 3. Please use only one type of representation for the same results.
line 492-493: unclear sentence
Round 2
Reviewer 2 Report
The authors provided a reviewed version of the initial manuscript and addressed well the majority of my comments. However there are still some issues which they need to be resolved before the manuscript can be considered for publication. The authors are encouraged to invest more time on dealing with the following comments:
- Citations not in the correct order. From [13] in line 62 they go to [32] in line 66. The newly added citations should be sequentially numbered and all citations must be renumbered.
- Table 1 should be reformatted. The parameters should appear in columns.
- Line 120: How do you know that the model can be used with hyperspectral imagery? Please justify or remove.
- Figures 1 & 2 should be merged in one figure. Figures 2a and 2b can be overlaid and fig.1 can be minimized and used as legend.
- Line 175. How the orthorectification was implemented?
- Paragraph 2.3.4: please mention if this approach for screening outlier depth values has been implemented before and cite if necessary. Otherwise mention that this is your own approach.
- Response 7 (to my previous version comment) should appear inside the text.
- In paragraph 4.1 it is not explained how the A and B areas in Figure 6 are produced. Please describe within the text, how did you make the map shown in figure 6?
- Lines 529-544 should go to the Discussion section and they should be justified in further detail.
- References not formatted in the journal style.
- Extensive English corrections are required.
